# Effectiveness and experiences with differentiated service delivery of HIV care in Kisumu County, Kenya: A mixed methods study, 2014–2021

Francesca Odhiambo[1]*, Raphael Onyango Mando[1], Jayne Lewis-Kulzer[2], A. Rain Mocello[2], Maurice Aluda[1], Edwin Mulwa[1], Appolonia Aoko[3], Paul Musingila[3], Elizabeth Bukusi[1], Craig R. Cohen[2]

**1** Centre for Microbiology Research, Research Care and Training Program, Kenya Medical Research Institute, Nairobi, Kenya, **2** Department of Obstetrics, Gynecology & Reproductive Sciences, University of California San Francisco, San Francisco, United States of America, **3** Division of HIV & TB, Global Health Center, US Centers for Disease Control and Prevention, Kisumu, Kenya

* fodhiambo@kemri-rctp.org

## Abstract

The adoption of the test and treat policy by the World Health Organization (WHO) in 2015 led to an unprecedented increase in the number of people living with HIV (PLHIV) enrolling into HIV treatment, thereby increasing the burden on HIV service delivery. To compensate, WHO endorsed the Differentiated Service Delivery (DSD) approach to reduce the burden on the health care system and therefore support attainment of the UNAIDS 95-95-95 goals by 2030. This study examined clinical outcomes among clients enrolled in the DSD models and examined health care worker and client experiences of the DSD approach. A client-level pre-post study was conducted in 14 Ministry of Health (MOH) facilities in Kisumu County from October 2014 – March 2021 to examine retention and viral load suppression (<1000 copies/ mL) in a cohort of stable clients aged 20 years and above at three time points: immediately preceding DSD start (pre-DSD; 2014–2016), 12 months post-DSD implementation (midline), and 24 months post-DSD (endline). Focus group discussions (FGDs) were conducted to assess DSD experiences among a sample of adult clients and health care workers. Findings from the pre-post analysis showed a significant increase in retention at 12 months (99.2%) and 24 months (98.9%) compared to pre-DSD (86.4%; p<0.001). The predominant themes shared by clients and healthcare workers in FGDs were high satisfaction with DSD due to the efficiency of services, improved staff attitudes, and reduced clinic workload. Clients also expressed a strong preference for facility-based models owing to perceived stigma and privacy concerns associated with community DSD models. This study provides important insights on the promising effectiveness of DSD models on sustained retention on ART and viral suppression and the acceptability of this modality for client-centered HIV care.

**Data availability statement:** All relevant data for this study are publicly available from the Zenodo repository (https://doi.org/10.5281/zenodo.15491841).

**Funding:** This study was financially supported by the President's Emergency Plan for AIDS Relief (PEPFAR) through the Centers for Disease Control and Prevention (CDC) under the terms of Cooperative Agreement (NU2GGH001947) received by EB and CRC. The findings and conclusions in this report are those of the authors and do not necessarily represent the official position of the funding agencies. No additional external funding was received for this study. The funder had no role in study design, data collection and analysis, decision to publish, or preparation of the manuscript.

**Competing interests:** The authors have declared that no competing interests exist.

## Background

HIV care and treatment has undergone major changes in recent years with the dual roll out of universal test and treat for everyone living with HIV and differentiated service delivery (DSD) models. The test and treat approach recommended by World Health Organization (WHO) has escalated the number of people living with HIV (PLHIV) receiving antiretroviral treatment (ART) [1–4], with ART coverage for PLHIV increasing globally from 46% in 2015 to 77% in 2023 [5,6]. The adoption of the test and treat policy also increased the burden of HIV service delivery to already con-strained health facilities, worsening clinic congestion, including in Kenya where the test and treat policy rolled-out in 2016 [7–12]. In response to the increasing number of PLHIV on ART, WHO endorsed DSD to reduce the health care burden through a sim-plified, client-centered approach across the HIV cascade. DSD was also developed to accelerate attainment of HIV epidemic control, hence facilitating achievement of the Joint United Nations Programme on AIDS (UNAIDS) 95-95-95 goals by 2030 [13].

With the roll-out of DSD, it is essential to ensure that quality services are delivered effectively and efficiently to sustain or improve clinical outcomes including retention in HIV care and viral load suppression. Poor retention in HIV care and treatment pro-grams has been identified as the most common reason for treatment failure among PLHIV on ART [14,15]. In Kenya, retention in care is estimated to have dropped to 69% by 36 months following ART initiation among adults [16,17]. Retention in care is tied to viral suppression and is central to attainment of UNAIDS 95-95-95 goals for epidemic control through reduction in community viral load [5,12,13,18]. Several stud-ies in sub-Saharan Africa have reported comparable effectiveness of DSD models to standard facility-based care in sustaining high retention rates in HIV care and viral suppression [19–21]. However, there are variations in study designs, DSD models evaluated, populations, and outcome measurements.

Kenya adopted the test and treat strategy in 2016 and initiated the roll-out of DSD approaches in early 2017 [12,22,23]. In Kenya, DSD models have been evaluated in the context of continuation of services during the COVID-19 pandemic [24], as well as client preferences and factors associated with enrollment into different DSD models [25,26]. Yet, there is a need to examine the effects of DSD implementation on clinical outcomes in the Kenyan context.

Family AIDS Care & Education Services (FACES), a U.S. President's Emergency Plan for AIDS Relief (PEPFAR)/ U.S. Center for Disease Control (U.S. CDC) implementing part-ner, supported the Kisumu County Ministry of Health (MOH) to implement its DSD model in Kisumu County, Kenya. This study examined the effectiveness of DSD approaches at improving and maintaining client retention and clinical health outcomes, including viral suppression, and explored client and health care worker experiences with DSD.

## Methods

### Study design

We conducted a mixed-methods study in Kisumu, Kenya. The quantitative compo-nent of the study relied on data abstracted between 01/10/2014 to 30/09/2020 and

the qualitative component was conducted in August 2020. The study comprised a pre-post analysis of routine clinical data to assess the impact of DSD on retention in HIV care and HIV viral suppression among PLHIV. The qualitative component included FGDs with health care workers and clients at a subset of health facilities to gain a deeper understanding of their experiences and perspectives with DSD.

## Study setting

This study was conducted in 14 MOH facilities in Kisumu County, Kenya where the HIV prevalence is 17.5% compared to the national average of 4.9% [27]. FACES, a collaboration of the UCSF and KEMRI supported HIV services in these health facilities [28].

## Participants

Participants for the pre-post study on retention in care and viral load suppression were drawn from health facilities with electronic medical records (EMR) systems. Data were included for PLHIV, 20 years or older, who were stable in HIV care throughout the pre-DSD period of October 2014-October 2016. We defined "stable" as per the national DSD guidelines and included clients who had been on their current ART regimen for at least 12 months; were virally suppressed (<1,000 copies/ml) at their most recent viral load within the prior 12 months; were adherent to scheduled clinic visits; were not pregnant or breastfeeding between October 2014-October 2016; and did not have opportunistic infections and tuberculosis in the prior 6 months. Clients left the cohort if they dropped out of care, transferred to another health facility, died, or became unstable in clinical care. Clients entered the cohort by transferring-in from another facility and/or meeting the inclusion criteria for stability (20). New entrants entered the cohort at any time once they had achieved the mandatory 12-month maturity period. The clients included in this cohort were followed through September 2020.

For the qualitative component, we conducted a total of nine FGDs, each having 10 participants derived from 3 of the 14 health facilities in this study. At each of the three facilities, we held 3 FGDs that included: a) health care worker group, b) adult male client group, and c) adult female client group. We purposively sampled to ensure that each group included 8–10 participants representing a variety of client interactions for facility-based health care workers and multiple differentiated care options for the client FGD sessions.

## Intervention approach

In 2017, FACES rolled-out differentiated services in accordance with the national DSD guidelines across all supported health facilities. Facility and community-based DSD models were introduced, both allowing refills every three months and health facility visits for clinical review every six months, along with decentralization to dispensaries, task shifting to nurse visits, and community ART distribution points [10,22,23]. Models also included rapid facility drug pick-up (FastTrack), community-based ART groups (CAGs), and facility-based ART groups (FB-AGs). Technical teams used National AIDS and STIs Control Program (NASCOP) tools and guidelines to train, sensitize, and equip health facilities to effectively provide differentiated care services.

## Measures

We defined exposure as receiving DSD at the time DSD was implemented at each facility ("baseline"), as recorded in the medical record. Our outcome of interest was retention in DSD, defined as being active in care (clinic attendance) during the last 12 (11–13-month window) and 24 months (23 – 25-month window) since DSD start, and HIV viral suppression, defined as <1000 copies/mL. To establish a baseline DSD that mirrored DSD inclusion criteria prior to DSD roll out, we calculated the proportion of clients retained in care and virally suppressed 12 months prior to DSD start and those who were adherent to scheduled clinic visits and not pregnant or breastfeeding between October 2014-October 2016

("baseline") and compared the baseline retention to retention in care at 12 months post-DSD start ("midline"), and at 24 months post-DSD start ("endline"). Similarly, we calculated the proportion of clients with viral suppression at midline and endline to examine outcomes. We extracted viral load data for the period October 2015-September 2016 for the baseline and between October 2017 and September 2018 for the midline period and October 2020 to September 2021 for the end line period. We analyzed viral load suppression data only for clients that had viral load done. The data for the viral load data analysis used here was sourced separately from the national viral load website (https://nascop.org) and so viral load data includes only those with some viral load outcome.

## Data collection

Client demographic and clinical data were collected on standardized MOH clinical forms. The data was entered into Open Medical Record System (OpenMRS) and Kenya Electronic Medical Record system (KenyaEMR), the two EMR systems used at the FACES-supported facilities. Data were extracted using customized Structured Query Language (SQL) queries for the period of 01/10/2014 to 30/09/2020. The IRB waived the requirement for informed consent for the data from the patients enrolled in routine care. Our data is hosted on Zenodo [29].

Weekly data quality assessments were conducted to address misclassifications and data gaps raised by the data management and analysis team. Client DSD model was classified based on inclusion criteria and documentation of their DSD model in the EMR system. If the model was not documented or conflicted with clinic refill/visit history, we used inclusion criteria and refill/clinic visit schedules to deduce those receiving differentiated care.

Qualitative data collection was carried out in August 2020. The informed consent process and FGD sessions took place in a private room at the health facility. The client FGD participants were assigned numbers for use during the FGD to protect confidentiality. FGDs among health care workers were conducted in English while client FGDs were conducted in the language preferred by the group (English, Kiswahili, or Dholuo). A trained facilitator led the sessions using semi-structured guides with a note-taker present and audio-recording to document the discussions, with each FGD taking about 90 minutes. The discussion guides focused on differentiated care experiences, challenges, benefits, and recommendations for improvement (S1 and S2 Files).

## Data analysis

We used summary statistics to describe baseline demographic and clinical characteristics of clients by model of care delivery. We assessed bivariate relationships between sociodemographic characteristics (age groups, sex, Kenya Essential Package of Health (KEPH) level, and urbanity. KEPH is the categorization of packages of services provided by health facilities by level of facility.) -and overall retention at baseline, 12 months (midline), and 24 months (endline) post-DSD implementation. We further used bivariate analysis to assess retention over time by model of care. We performed relative risk analysis to calculate the probability of retention at midline and endline in reference to the baseline, reporting risk ratios at 95% confidence intervals (CIs) for retention at baseline, midline, and endline, for the different demographic and clinical characteristics. In addition, we analyzed the data for the clients imputed to be in DSD to support the decision to collapse that group with those noted explicitly as being in DSD. We conducted analysis to examine clinical encounters for any instance of 180+/-14 days apart between any clinical encounters. Quantitative analyses were conducted in STATA 16.1 (StataCorp, College Station, TX).

The qualitative audio-recordings were transcribed and translated to English, if the local language was used, and stored securely on a password protected folder. To assure for transcription quality, an experienced qualitative study staff member validated the accuracy of the FGD transcript translation and content by directly comparing the audio file to the transcription file. Any changes needed were discussed and made jointly by the two-person qualitative study team. Deductive and inductive approaches were used for codebook development based on themes in the guide and insights gained from the transcripts. The codebook and transcripts were uploaded to Dedoose analytic software (SocioCultural Research

Consultants, Manhattan Beach, CA) for coding by the qualitative study team. To achieve inter-coder reliability, the coding team coded two FGD transcripts jointly and discussed the differences in the application of codes to build consensus and consistency in the application of codes. The remaining transcripts were divided for individual coding. A thematic analysis was carried out to explore and identify key themes and patterns on the DSD experiences, challenges and recommendations in the context HIV care.

### Ethical considerations

This study was reviewed and approved by the US Centers for Disease Control and Prevention (U.S CDC), the Kenya Medical Research Institute (KEMRI) Scientific Ethics Regulatory Unit (SERU; NPR1/2009) and University of California San Francisco (UCSF) Institutional Review Board (#11–05348). A waiver of consent was obtained for the clinical data since it was captured as part of routine care. Written informed consent was obtained prior to FGD client and HCW participation.

§ See 45 C.F.R. part 46.101(c); 21 C.F.R. part 56.

### Results

Of the 43,304 clients assessed during the pre-DSD period (Fig 1), 24,296 clients met the stability criteria and were included in the analysis of retention in the pre-DSD period. The 12-month retention was 86.4% at baseline.

At baseline, 15,747 clients, 65% female (Table 1), met the eligibility criteria for stable status and therefore were eligible for DSD (Fig 2); 11,402 (72.4%) of these clients elected to receive care through a DSD model, and the remaining 4,345 (27.6%) elected to continue with standard of care (SOC). A higher proportion of SOC clients compared to those who elected a DSD model were in the youngest age category (20–34 years of age; 46.1% vs. 29.4%, respectively) and had a lower WHO stage at HIV diagnosis (WHO Stage 1; 66.3% vs. 39.3%, respectively).

In comparison to the pre-DSD period, overall clinical retention improved significantly for clients at midline (86.4% vs.99.2%, p<0.001; Table 2) and endline (86.4% vs.98.9%, p<0.001) evaluation periods. Outcomes were similar across age groups and sex. Among the 11,402 clients enrolled in a differentiated care model, 99.2% were retained at midline. Of those enrolled, 4,630 had completed 24 months of follow-up at the time of analysis, and 98.9% were retained on treatment at endline. These trends persisted when stratified by sex and age. Nearly all clients in DSD at baseline were enrolled in the FastTrack ART/Express model (11,026, 97.1%), 99.6% of whom were retained in care at the midline and 99.0% at endline. Clients in Community ART Groups and Facility-based ART Groups also achieved high rates of retention at midline (100% and 95.2%, respectively) and endline (100% and 95.7%, respectively). Retention improved universally across sex, age, and DSD model in comparison to pre-DSD (Table 2). When conducting sensitivity analysis to examine clinical encounters for any instance of 180+/-14 days apart between any clinical encounters, of 6,169 clients, we found 3024 (49%) who would be in other packages of care but were assigned to SOC.

Viral suppression improved significantly at the midline (93.9%; p<0.001) and endline (95.7%; p<0.001) post-DSD periods compared to baseline (90.9%) (Table 3). HIV viral suppression rates were similar across sex and age groups at all evaluation periods.

In relative risk analysis (S1 Table), males were 1.015 times likely to be retained at midline than at baseline (R.R 1.015, 95% C.I [1.005-1.026]), and 1.152 at endline in reference to baseline (R.R 1.152, 95% C.I. [1.141–1.163]). Females were 1.138 times likely to be retained at midline in reference to baseline (R.R 1.138, 95% C.I. [1.131–1.145]), and at endline in reference to baseline (R.R 1.138, 95% C.I. [1.130–1.146]). Age; clients aged 20–34 years were 1.2 times likely to be retained at midline in reference to baseline (R.R 1.211, 95% C.I. [1.202–1.221]); 35–49 year olds were 1.09 times likely to be retained at midline in reference to baseline (R.R 1.087, 95% C.I.[1.079–1.094]), and at endline were also 1.09 tiles likely to be retained at endline in reference to baseline (R.R. 1.087, 95% C.I.[1.079–1.095]); among 50+ year olds the likelihood of retention at midline was 1.07 times likely to be retained at

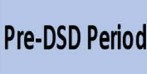

**Pre-DSD Period**

**Assessed for eligibility (n = 43,304) January- September 2015***
*May be interpreted differently at each facility based on DSD Start Date

**Not advanced to baseline analysis (n = 19,008, 43.9%)**
*Below 20 years (n = 8,713, 45.8%)
*Breastfeeding (n = 2, <0.01%)
*Not virally suppressed (n = 5,605, 29.5%)
*< 12 months in current ART regimen (n = 104, 0.01%)
*Having TB (n = 554, 2.9%)
*Pregnancy (n = 957, 5.0%)
*Having Opportunistic infections. (n = 366, 1.9%)
*Absent additional visits (n = 1,709, 9.0%)
*Not yet matured in care (n = 998, 5.3%)

**Pre-DSD**

**Stable clients 12 mo. prior to DSD**
(n = 24,296, 56.1%)

**Not Retained (n = 3,291, 13.7%)**
LTFU (n = 3,179, 96.6%)
Dead (n = 112, 3.4%)
Transfer Out (n = 38, 1.2%) *

**Retained (n = 20,967, 86.3%)**

*Transfer outs are excluded from the denominator and numerator*

**Fig 1. Pre-DSD flow chartg of Assessment of Clients Eligible for DSD.**

midline in reference to baseline (R.R 1.073, 95% C.I. [1.062–1.085]), at endline they were 1.066 times likely to be retained in reference to baseline (R.R 1.066, 95% C.I. [1.053–1.078]). Similarly, WHO Stage 1 was 1.173 times likely to be retained at midline in reference to baseline, (R.R 1.173, 95% C.I. [1.164–1.183]) while at endline was 1.007 times likely to be retained in reference to baseline, (R.R 1.007, 95% C.I. [1.002–1.013]). Those in WHO Stage 2 were 1.1 times likely to be retained at midline in reference to baseline (R.R 1.10, 95% C.I. [1.092–1.109]) and 1.093

**Table 1. Demographic and clinical characteristics stratified by model of care delivery at baseline, Kisumu County, 2014-2021.**

| Total | | | Standard of Care | | DSD | |
|---|---|---|---|---|---|---|
| **Variables** | N | % | N | % | N | % |
| **Total** | 15,747 | 52.7 | 4,345 | 14.5 | 11,402 | 38.2 |
| **Sex** | | | | | | |
| Male | 5,511 | 35.0 | 1390 | 32.0 | 4121 | 36.1 |
| Female | 10,236 | 65.0 | 2955 | 68.0 | 7281 | 63.9 |
| **Age** | | | | | | |
| 20-34 years | 5,352 | 34.0 | 2,001 | 46.1 | 3,351 | 29.4 |
| 35-49 years | 7,289 | 46.3 | 1,693 | 39.0 | 5,596 | 49.1 |
| 50+years | 3,106 | 19.7 | 651 | 15.0 | 2,455 | 21.5 |
| **Kenya Essential Package of Health Level** | | | | | | |
| Level 2 | 2,165 | 13.7 | 690 | 15.9 | 1,475 | 12.9 |
| Level 3 | 2,272 | 14.4 | 912 | 21.0 | 1,360 | 11.9 |
| Level 4 | 11,310 | 71.8 | 2,743 | 63.1 | 8,567 | 75.1 |
| **Sub County** | | | | | | |
| **Urban** | | | | | | |
| Kisumu East West Central | 12,543 | 79.7 | 3,408 | 78.4 | 9,135 | 80.1 |
| **Rural** | | | | | | |
| Muhoroni | 273 | 1.7 | 11 | 0.3 | 262 | 2.3 |
| Nyakach | 699 | 4.4 | 65 | 1.5 | 634 | 5.6 |
| Nyando | 2,232 | 14.2 | 861 | 19.8 | 1,371 | 12.0 |
| **Components of the package of ART distribution options** | | | | | | |
| Community ART Groups [CAGs - 8 sites] | 230 | 1.5 | – | – | 230 | 2.0 |
| FastTrack ART/Express- 14 sites | 11,108 | 70.5 | – | – | 11,108 | 97.4 |
| Facility-based ART Groups [FB-AG]- 8 sites | 64 | 0.4 | – | – | 64 | 0.6 |
| Standard Care | 4,345 | 27.6 | 4,345 | 100.0 | – | – |
| **WHO Staging Start of Intervention** | | | | | | |
| WHO Stage 1 | 7,364 | 46.8 | 2,880 | 66.3 | 4,484 | 39.3 |
| WHO Stage 2 | 4,593 | 29.2 | 841 | 19.4 | 3,752 | 32.9 |
| WHO Stage 3 | 3,129 | 19.9 | 531 | 12.2 | 2,598 | 22.8 |
| WHO Stage 4 | 640 | 4.1 | 77 | 1.8 | 563 | 4.9 |
| Missing | 21 | 0.1 | 16 | 0.4 | 5 | 0.04 |

times likely to be retained at endline in reference to baseline (R.R 1.093, 95% C.I. [1.083–1.103]). Clients in WHO Stage 3 were 1.09 times likely to be retained at midline in reference to baseline (R.R 1.085, 95% C.I. [1.075–1.094]) and endline were 1.08 times likely to be retained in reference to baseline (R.R 1.081, 95% C.I. [1.071–1.091]). Clients on WHO Stage 4 were 1.07 times likely to be retained at midline in reference to baseline (R.R 1.068, 95% C.I. [1.047–1.089]) while at endline they were 1.07 times likely to be retained in reference to baseline (R.R 1.065, 95% C.I [1.04–1.09]. See more details in S1 Table.

## Qualitative findings: health care worker and client experiences with DSD

The DSD qualitative FGD findings from 9 FGDs, 6 with adult clients (3 male and 3 female FGDs) and 3 with health care workers including facility-based nurses, clinical officers, pharmacy and laboratory technologists, data clerks, and lay health care workers are summarized in the themes below along with illustrative quotes.

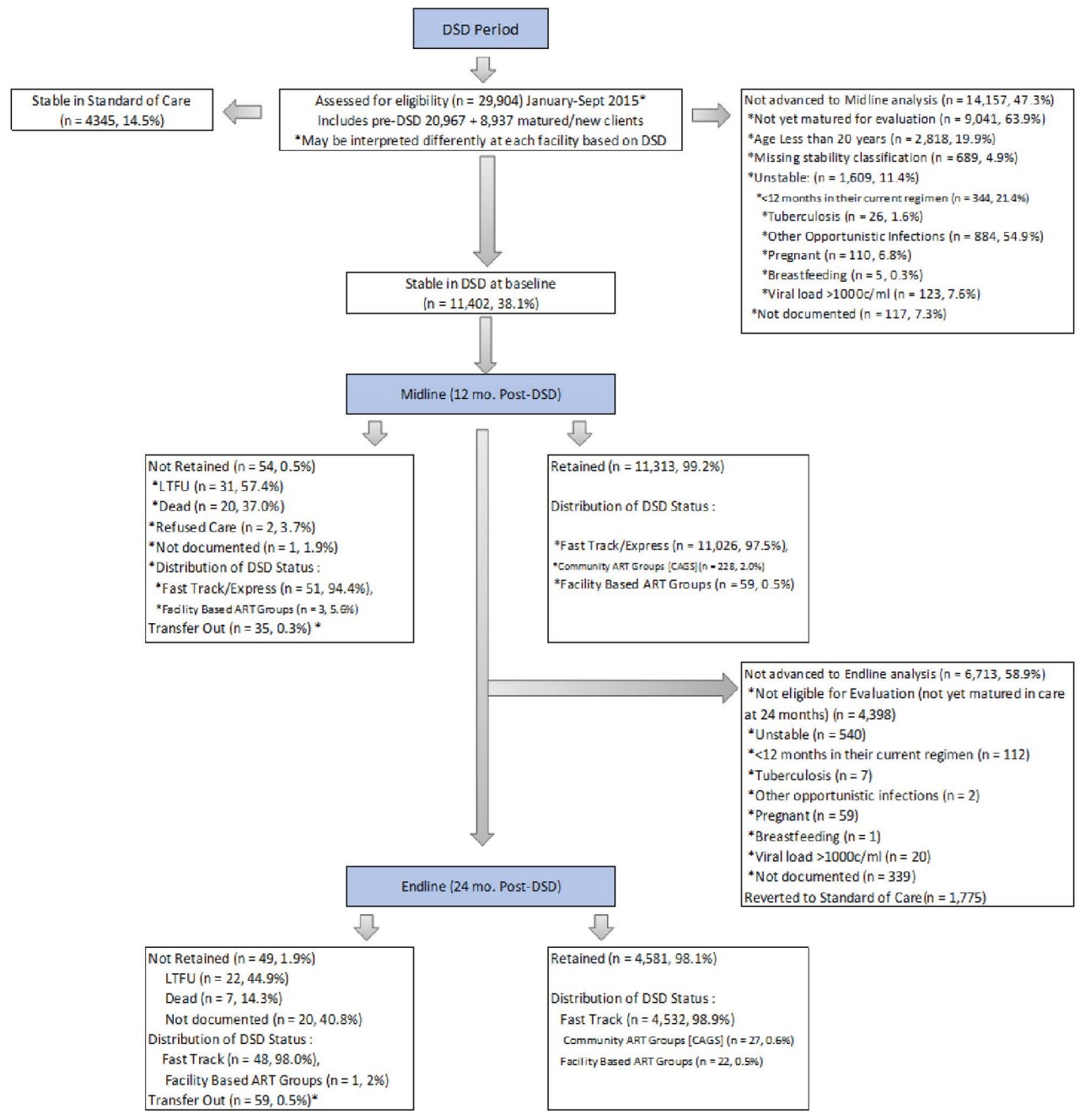

*Transfer outs are excluded from the denominator and numerator

**\* Reflects all clients in care, based on routine clinical encounters.**

**Fig 2. DSD Flow Chart of Assessment of Clients Eligible for DSD at Month 12 and Month 24.**

**Table 2. Retention at midline and endline compared to baseline, by model of care and demographic characteristics among all stable clients, Kisumu County, 2014-2021.**

| | Baseline | Midline | Endline |
|---|---|---|---|
| | N (%) | N (%) | N (%) |
| Overall Retention | 20,967 (86.4) | 15,586 (99.2) | 6632 (98.8) |
| Gender | | | |
| Male | 7,513 (85.6) | 5,461 (99.1) | 2336 (98.6) |
| Female | 13,454 (86.9) | 10125 (98.9) | 4296 (98.9) |
| Total | 20, 967 (86.4) | 15,586 (99.2) | 6,632 (98.8) |
| Age | | | |
| 20–34 years | 9,920 (81.4) | 5275 (98.6) | 2381 (98.7) |
| 35–49 years | 8,150 (91.2) | 7227 (99.1) | 3014 (99.1) |
| 50+years | 2,897 (92.5) | 3084 (99.3) | 1237 (98.6) |
| Total | 20,967 (86.4) | 15,586 (99.2) | 6,632 (98.8) |
| Components of the package of ART distribution options | | | |
| Community ART Groups [CAGs] | n/a | 228 (100) | 27 (100) |
| FastTrack ART/Express | n/a | 11,026 (99.6) | 4,532 (99.0) |
| Facility-based ART Groups [FB-AG] | n/a | 59 (95.2) | 22 (95.7) |
| Standard of Care [SOC] | 20,967 (86.4) | 4,273 (98.3) | 2,051(98.5) |
| Total | 20,967 (86.4) | 15,586 (99.2) | 6,632 (98.8) |

Chi square test of proportions reporting Fisher's exact test* All p-values significant at $p < 0.05$.

## Efficiency of services

Among clients and health care workers, the predominant emerging theme surrounding the benefits of DSD was high satisfaction with the efficiency of services in the clinic. Clients appreciated spending less time at the clinic and having more time to focus on other personal responsibilities. Health care workers appreciated the reduced workload, less congested facilities, and ability to have more focused time for clients with clinical care needs.

> *Client-Female: "There has been a great change, often a single mother like me could come to the clinic, stay up too late in the evening then because I rely on a day-to-day work and pay, I end up sleeping without food because the entire day had been spent at the clinic. So, this fact that when we come things are easier and faster taking very little time then going back to our work makes me happy."*

> *Client: "I am a fisherman; I sometimes move to far places in search of fish and the three months refill gives me an opportunity to concentrate on my work for a longer period and when time comes for refill it is easy to come and pick and get back to my work immediately without any delay."*

> *Health care worker: Okay, I think differentiated care has really helped in the essence that the congestion at the facility has reduced and it minimizes the time [clients take] for coming to the facility. It also gives us health care workers time so that we [can] attend to the individual patient who has come with a problem without being on a rush…"*

## Staff attitudes

Both clients and health care workers noted improved staff attitudes and more meaningful client encounters with the reduced workload.

**Table 3. Program-level viral load suppression among clients before and after implementation of differentiated care among clients aged ≥20 years, Kisumu County, 2014-2021.**

| Characteristics | Pre-DSD | Midline* | Endline* |
|---|---|---|---|
| | Viral load <1,000 c/ml | Viral load <1,000 c/ml | Viral load <1,000 c/ml |
| | N (%) | N (%) | N (%) |
| **Overall** | 31,124 (90.9%) | 36,492 (93.9%) | 44,430 (95.7%) |
| **Sex** | | | |
| Female | 21,108 (91.2%) | 24,960 (94.4%) | 29,935 (96.2%) |
| Male | 9,896 (90.9%) | 11,504 (93.1%) | 14,494 (95.0%) |
| Total* | 31,004 (90.5%) | 36,464 (93.8%) | 44, 429 (95.7%) |
| **Age (years)** | | | |
| 20–34 years | 11,821 (89.3%) | 13,628 (93.2%) | 15,381 (95.1%) |
| 35–49 years | 13,113 (92.0%) | 15,508 (94.3%) | 18,965 (96.1%) |
| 50＋years | 6,190 (92.7%) | 7,312 (94.6%) | 10,084 (96.2%) |
| Total* | 31,124 (90.9%) | 36,448 (93.8%) | 44,430 (95.7%) |
| **Sub county** | | | |
| Nyando-Kisumu East West and Central | 20,071 (93.6%) | 22,982 (95.2%) | 25,422 (97.0%) |
| Muhoroni | 5,405 (87.7%) | 6,608 (91.7%) | 8,951 (92.8%) |
| Nyakach | 5,648 (86.2%) | 6,902 (92.2%) | 10,057 (95.5%) |
| Total | 31,124 (90.9%) | 36,492 (93.9%) | 44,430(95.7%) |
| **KEPH Level** | | | |
| Level 2 | 5,225 (88.2%) | 6,422 (93.2%) | 8,715 (95.2%) |
| Level 3 | 4,624 (89.0%) | 5,707 (92.4%) | 7,390 (95.0%) |
| Level 4 | 21,275 (92.3%) | 24,363 (94.5%) | 28,325 (96.2%) |
| Total | 31,124 (90.9%) | 36,492 (93.9%) | 44,430(95.7%) |

Comparison of midline and endline to pre-DSD, all p-value<0.001 using chi square test of proportions reporting Fisher's exact test at 95% CI *Totals may not add to overall because missing data points of sex and age are excluded.

*Client: "The change I have noticed is that when you come for the six-monthly clinical visits the clinician takes much time with the patient compared to when I was coming on a monthly visit."*

*Health care worker: "We are having a happier workforce because the psychological challenge as a result of being fatigued all the time is a thing of the past."*

## Perceived stigma and privacy

Perceived stigma with community models was a common thread among both clients and health care workers due to privacy and confidentiality concerns. As a result, facility-based models were overwhelmingly preferred by clients.

*Client: "I prefer the facility FastTrack model because the community- based ART groups are prone to stigmatization."*

*Health care worker: "FastTrack. In FastTrack, the client's privacy is guaranteed for them in the facility."*

*Health care worker: "You know, once you get there and I see you with that envelope, and that blue plate motorbike [government vehicle], automatically I will know that you are taking ART."*

## Care motivation

Clients and health care workers agreed that DSD services added to clients' motivation to adhere and stay virally suppressed as they did not want to lose the privilege of DSD.

> Client: "…the motivation of having to come after three months for refill and six months for clinical visits have made me put more effort into adhering to medication so that I can continue with differentiated care services."

> Health care worker: "The impact is positive, in the sense that the clients who are in the FastTrack or given any form of the differentiated model of care, … it makes them work harder on the adherence issues so that they maintain the same model of care."

> Health care worker: "Okay, for the people who are taken for sample for viral load, when they come back for a refill, they are very much concerned about their viral load and if you tell them that they are virally suppressed, they are very happy and would wish to maintain their DSD model."

## Autonomy

Clients revealed satisfaction with having more autonomy over their clinical care, felt less stigmatized and more normalized, since they were no longer spending a lot of time at the clinic.

> Client: "I don't come to the clinic frequently. People were used to seeing me leave my home for the clinic on a monthly basis but now they have even forgotten that I visit the clinic and can't notice to stigmatize me when I visit. They now see me as equal to them because I am healthy, alive, and going about my business just like all of them without any interruptions."

> Health care worker: "Yeah, like taking it positively like they are just like any other person. …they are not coming to the hospital … every other time. So, it is normal, they are taking it like a normal lifestyle now."

## Recommendations from the focus group discussions

Recommendations centered on further spacing of refill and clinical visits, improved privacy measures for discrete community delivery, more client education on DSD services to promote adherence and suppression. Staggered clinic hours with staff incentives to accommodate clients who needed to come early or late was also recommended.

Clinic and refill spacing

> Client: "I … wish that it [differentiated care] could be changed to yearly appointment. If you don't have any conditions or illness and you are stable I don't see anything wrong with having clinical reviews after 12 months. We are advised to visit the clinic any day any time whenever we feel sick, so with that in mind, I think it is possible."

> Health Care Worker: "I think one of the services that they [clients] will really need is the longer TCAs."

> Health Care Worker: "I think instead of giving clients three months drugs and then coming back for refill, I would prefer we give them the drugs for six months especially for this person who comes from a bit far to reduce the costs attached in travelling."

Privacy

> Health care worker: "…clients become stigmatized or fear of taking drugs because it seems like they are meeting in an open area, they are meeting somewhere everyone will come to see what they are doing. Transparent containers

*and the brown envelopes contribute to stigma. If we could provide something of that sort [big bag], it would be better because no one knows whatever you are carrying and if you are going somewhere like a house no one knows what is going to take place there."*

Patient education

*Health Care Worker: "I think they [clinic] should also have regular consistent patient education so that the patients are aware in as much as you are on DFC [differentiated care], it does not mean that you are now not supposed to come for your clinic visits and follow-ups… they [clients] have also to take the initiative of their own health."*

Staggered clinic hours with staff incentives

*Health Care Worker: "…those [clients] like the working class who always come as early as seven. Some health workers may come earlier than working hours and some may extend past official time. At times those clients call, they want to come after five, maybe for the benefit of clients and for the staff who come early and leave early, an extraneous allowance or something related to that."*

## Discussion

This study found that clients in both DSD and SOC had improved retention in HIV care over time compared to baseline, with those in enrolled in DSD having even higher rates of improved retention and viral suppression compared to baseline. This improvement was observed across age groups, sex, Kenya Essential Package of Health (KEPH) level, and urbanity. Several studies in sub-Saharan Africa have found sustained retention and viral suppression among individuals receiving DSD over time and similar findings among those in SOC including a meta-analysis indicating retention in care within about 5% between DSD and SOC and some improvement viral suppression among those in DSD [20,30–32]. A study in Kenya with stable clients assigned to a differential care model across 15 clinics found a significant association with differentiated care and longer retention in care [21]. A meta-analysis among adolescents and young adults living with HIV also found that differentiated care significantly increases rates of retention and viral suppression [33]. Although our study found higher retention and viral suppression rates among the cohort in DSD, the increase may in part due to the cohort being clients who were well-established and stable in care. The high viral load suppression across the program may be explained, in part, by the introduction of the integrase inhibitor dolutegravir (DTG) as the preferred first-line regimen for all adults and adolescents with HIV. Kenya was among the first countries to adopt the WHO recommendations and rapidly transitioned and/or initiated all eligible PLHIV to DTG-based regimens from 2019. These collective findings illustrate promising sustained clinical health outcomes among clients living with HIV receiving streamlined differentiated care services.

As illustrated in the qualitative findings, clients, and health care workers alike preferred less frequent clinic visits for the reduced burden to clients, staff, and health facilities. They appreciated the efficiencies of DSD - recognizing that less frequent visits translate into fewer costly and time-consuming trips to the clinic, reduced congestion and waiting time, improved staff attitudes, and having more time for other daily responsibilities. DSD appears to tackle multiple HIV care obstacles that clients face, thereby facilitating healthy behavior by making it easier and less demanding of them. On the health care worker side, more time focused on managing complex cases and bolstering quality of care was likely not only a benefit to clients enrolled in DSD models but also a benefit to the health care workers who felt less rushed and fatigued, had better attitudes, and more time to manage clients. Other research has shown that when retention strategies are employed and barriers addressed, good treatment adherence is sustained [34–38]. Clients expressed interest in expanding DSD by increasing the duration of ART refills from 3 months to 6 months with annual clinical review, a concept that has been tried at a smaller scale in similar settings with clients reporting a greater sense of personal freedom and normalcy

[39]. However, low rates of return for annual clinical reviews and completion of viral load monitoring, and lower viral load suppression for clients receiving 6 monthly ART dispensing with annual clinical review has been found [40]. Increased ART refill duration combined with strategies to increase adherence support through DSD models may be worth exploring.

This intervention yielded improvement over time among stable clients already highly engaged in care with good outcomes, potentially due to the way DSD may have augmented intrinsic motivation. As explained by the self-determination theory, intrinsic motivation drives individuals to embrace behaviors for their own internal satisfaction [41]. While PLHIV with optimal health outcomes may already have high levels of motivation to adhere to treatment for their health, their level of internal satisfaction may be further heightened by the benefits gained through DSD. During FGDs, both clients and health care workers indicated that participation in DSD indicated client status of well-being, and this motivated clients to sustain the convenience of DSD. DSD may therefore help increase healthy behaviors to sustain its benefits. This is supported by other research connecting optimal HIV care and treatment behaviors with internal benefits gained [41–43]. Client empowerment may also play a role in heightened retention and outcomes. DSD may give clients more autonomy to take responsibility for their own care. A study in Haiti showed increased decision-making among clients and perceived quality of care improvements motivated clients to support clinic system changes [44]. Clients in our study expressed feeling more trusted to adhere to their medication. This is also supported by the self-determination theory; when people have more autonomy and ability to manage their life, they feel more motivated and show enhanced performance [45]. Among those in the FastTrack model in this study, clients noted feeling more normalized about living with HIV and less stigmatized, which is associated with better outcomes [19,46].

Stable clients who remained in SOC also had increased retention outcomes. This may be explained by program-wide intensified retention strategies carried out during this evaluation period, beginning in 2018, for all clients: enhanced defaulter tracing, mHealth use, and weekly data reviews. Re-engaging defaulters included updating tracing algorithms to reach defaulting clients early with enhanced staff accountability. Defaulter management was streamlined through mHealth use with automated client reminders before a visit, along with tracing procedures for re-engagement if a visit was missed. Retention was further bolstered through review of facility-level dashboards, with weekly leadership review of retention data to identify gaps and institute improvements. The added benefits of DSD may have also played a role in SOC outcomes. For example, the reduced clinic workload, less crowded facility, and improved staff attitude, may have enhanced the clinical experience among those in SOC when accessing clinic reviews every 3 months.

This study found that most clients preferred the FastTrack model of care with the health facility as the preferred ART refill location. This is corroborated by other studies carried in sub-Saharan Africa [26,42,47,48]. In addition to FastTrack efficiencies, the protection of privacy and confidentiality is a major concern and more secure in the health facility than in the community. This is substantiated by a recent study carried out in Ghana [42]. Inadvertent disclosure of one's HIV status is a high risk, especially if community members recognize ART packaging and familiar courier and if they become curious about the community meeting points for ART pick up. These are important concerns to address and mitigate to sustain and increase patient participation in community models. Community-based clubs have been shown to be more acceptable as club locations are selected for their potential to maintain confidentiality [49]. Another benefit expressed by clients about the facility-based model is being able to access clinical care or advice when they come to the clinic for refills. Some clients did express the desire to have refill visits at the clinic harmonized with the facility-based adherence support group meetings, which would encourage more clients to attend for the added adherence support. A Discrete Choice Experiment in this setting revealed that adherence support at either the individual or group level was a preferred and important attribute of care for clients [26]. Facility-based groups could potentially be enhanced and expanded to accommodate this suggestion [42,43].

This study's strengths include utilizing a large sample size from a routine setting, patient level data and a long duration of follow-up to examine clinical outcomes in the DSD context. This study provides insights on the effectiveness of DSD models on sustained retention on ART and viral suppression. However, the findings in this study may be unique to this

study population and not necessarily represent a broader population of PLHIV. There may also be differences in the characteristics and timing of clients receiving various differentiated care model strategies due to the observational study design that lacked a control group. To address this issue of comparability in the analysis, we adjusted for clinical and demographic factors routinely captured in the EMR to control for any confounding by factors generally known (previously published) to be associated with outcomes of interest. Observed results may be biased if unmeasured confounders account for some or all the observed associations. During the time of differentiated care roll-out and this evaluation, program-wide intensified retention strategies were underway, which may have played a role in retention and viral load outcomes in both SOC and differentiated care groups, The introduction of DTG and subsequent transition of patients from other regimens may have also influenced high viral suppression findings. However, at baseline, only 0.01% of patients were on DTG and the number increased to 2.06% by 12 months (midline) and to 72.36% by 24 months (endline). Notably, at the time of this evaluation, Kenya national guidelines only targeted stable patients for DSD. DSD implementation has since expanded to include other patient populations, such as adolescents and pregnant women, which warrants evaluation of program outcomes in the context of this change. Within the qualitative component of this study, although we sought diversity in participants and health facility type, we did not obtain insights from those who elected not to participate in DSD and it is possible that opinions expressed in the FGDs may not be representative of DSD participants or health care workers in other settings.

## Conclusion

This study demonstrated that stable clients enrolled in DSD models yielded high retention in care and viral suppression. Clients and health care workers revealed high satisfaction with DSD due to more efficient services, improved staff attitude and reduced clinic workload. This study elucidated a preference for facility-based DSD models, due to perceived stigma and privacy concerns with community-based models and calls for exploring options for further spacing ART refill appointments. This evaluation provides important insights on how to improve and expand DSD services for optimal client-centered care and sustained outcomes.

## Supporting information

**S1 Table. Relative risk analysis of retention, overall and stratified by DSD track, sex, and age offsetting for clustering at facility level.**
(DOCX)

**S1 Checklist. STROBE Statement -checklist of items that should be included in reports of cross-sectional studies.**
(DOCX)

**S2 Checklist. Inclusivity in global research.**
(DOCX)

**S1 File. Differentiated care focus group discussion guide patient guide.**
(DOCX)

**S2 File. Differentiated care focus group discussion guide health care worker guide.**
(DOCX)

## Acknowledgments

The authors appreciate the participants for their valuable time, facility staff, KEMRI Director, and Family AIDS Care and Education Services (FACES) for their support of this EPMP evaluation.

## Author contributions

**Conceptualization:** Francesca Akoth Odhiambo, Raphael Onyango Mando, Jayne Lewis-Kulzer, Maurice Aluda.

**Data curation:** Francesca Akoth Odhiambo, Raphael Onyango Mando, A. Rain Mocello, Edwin Mulwa.

**Formal analysis:** Francesca Akoth Odhiambo, Raphael Onyango Mando, Jayne Lewis-Kulzer, A. Rain Mocello, Maurice Aluda, Edwin Mulwa, Paul K Musingila.

**Funding acquisition:** Elizabeth Anne Bukusi, Craig R. Cohen.

**Investigation:** Maurice Aluda, Craig R. Cohen.

**Methodology:** Raphael Onyango Mando, Paul K Musingila, Elizabeth Anne Bukusi, Craig R. Cohen.

**Project administration:** Francesca Akoth Odhiambo, Appolonia Aoko.

**Resources:** Appolonia Aoko, Elizabeth Anne Bukusi, Craig R. Cohen.

**Software:** Raphael Onyango Mando, Edwin Mulwa, Craig R. Cohen.

**Supervision:** Francesca Akoth Odhiambo, Jayne Lewis-Kulzer, Appolonia Aoko, Paul K Musingila.

**Validation:** Francesca Akoth Odhiambo, Jayne Lewis-Kulzer, A. Rain Mocello, Edwin Mulwa, Appolonia Aoko, Paul K Musingila, Elizabeth Anne Bukusi.

**Visualization:** Edwin Mulwa.

**Writing – original draft:** Francesca Akoth Odhiambo, Raphael Onyango Mando.

**Writing – review & editing:** Francesca Akoth Odhiambo, Raphael Onyango Mando, Jayne Lewis-Kulzer, Maurice Aluda, Appolonia Aoko, Craig R. Cohen.

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
