## [Decision Letter · Decision Letter 0]

PGPH-D-24-02789

Effectiveness and experiences with differentiated service delivery of HIV care in Kisumu County, Kenya: A mixed methods study, 2014-2021

Dear Dr. Mando,

Thank you for submitting your manuscript to PLOS Global Public Health. After careful consideration, we feel that it has merit but does not fully meet PLOS Global Public Health’s publication criteria as it currently stands. Therefore, we invite you to submit a revised version of the manuscript that addresses the points raised during the review process.

We look forward to receiving your revised manuscript.

Kind regards,

Joel Msafiri Francis, MD, MS, PhD

Academic Editor

Journal Requirements:

2. In the online submission form, you indicated that your data will be submitted to the Dryad database upon acceptance. Should your submission be accepted, we will require the following information in your Data Availability Statement:

a. The DOI provided by Dryad

b. The citation for your data package in the reference section of your manuscript

c. The citation for your data package in the methods section

If you are unable to adhere to our open data policy, please kindly revise your statement to explain your reasoning and we will seek the editor's input on an exemption.

Additional Editor Comments (if provided):

Reviewers' comments:

Reviewer's Responses to Questions

**Comments to the Author**

1. Does this manuscript meet PLOS Global Public Health’s publication criteria?

Reviewer #1: Yes

Reviewer #2: Partly

2. Has the statistical analysis been performed appropriately and rigorously?

Reviewer #1: Yes

Reviewer #2: No

3. Have the authors made all data underlying the findings in their manuscript fully available (please refer to the Data Availability Statement at the start of the manuscript PDF file)?

Reviewer #1: Yes

Reviewer #2: Yes

4. Is the manuscript presented in an intelligible fashion and written in standard English?

Reviewer #1: Yes

Reviewer #2: Yes

Reviewer #1: This is a really interesting study that described the 12-month and 24 month retention and viral suppression of adults >20 years enrolled in DSD. The qualitative component added richness to the data by giving insights into client perspectives of DSD and recommendations for improvement. The objectives are clear and the conclusions are supported by the data. Below are some areas for improvement

Introduction

The literature to back up the study rationale is lacking. There are many studies that describe clinical outcomes of DSD in adults living with HIV as well as client and HCW experiences. It would be useful to the reader to add some literature on clinical outcomes and client/ health care worker experiences from Sub-Saharan Africa and Kenya (if there are any), and then articulate the research gap more clearly.

Methods

Outcomes- why was a viral load cut off of <1000 copies/ml selected as the viral suppression outcome? Did the authors consider using an additional more stringent cut off such as <50 copies/ml in their analysis? Can the authors also elaborate more on the monitoring criteria used for viral suppression from the local Kenyan DSD guideline, and how this influenced the viral load cut off used in the study.

The authors need to clarify the time point that was used to define a 12-month viral load and a 24-month viral load. In clinical practice, viral loads are generally not taken at the specified time point. Can the authors clarify what the 12-month window and 24-month window was used for the viral suppression outcome.

Ethics

The authors state that a waiver was given not to obtain informed consent. Was this for the electronic data only? Why was no written informed consent obtained for the focus group discussions? Focus group discussions cannot be considered to be part of routine care.

Data collection

The description of the study methodology for the qualitative component is missing some important components. It is not clear what methodology was employed- thematic analysis, framework analysis, grounded theory. It is also not clear what approach was taken in analysing data, an inductive or deductive approach? How were coding discrepancies resolved? How was the accuracy of translation achieved for the non-English transcripts?

To enhance transparency and rigor of the manuscript, can the authors please provide a copy of the focus group interview guide used for this study as a supplementary file?

Data analysis

It is unclear how the exposure variables were selected. Can the authors elaborate more on how the exposure variables for the logistic regression model were selected?

The authors need to make it more clear how they calculated the retention post DSD enrolment, more specifically what exclusions were applied. From figure 2, it looks like those who did not make it to the 12 and 24-month follow-up were excluded from the analysis, but this is not articulated in the methods section.

Results

Line 203- Please add the percent symbol after 86.4.

Table 1, 2 and 3 are missing from the manuscript file. The results are referenced in the text but the tables are missing.

Line 226. The authors have given the viral suppression proportions at midline and endline. Did all clients have a viral load done? It seems unlikely that 100% of clients would have viral loads done at 12 and 24-month mark. What were the proportions for viral load completion at baseline, midline and endpoint? The denominator for viral load suppression would need to exclude those who did not have a viral load done.

Line 232-236. Please add the odds ratios that you are referring to in the text to make it easier for the reader to follow.

The authors have conducted a multivariable logistic regression analysis for retention at baseline, 12 months post DSD enrollment, and 24 months post DSD enrolment. The data on factors associated with viral suppression have not been presented. Can the authors consider adding this as part of their analysis as a supplementary table?

Discussion

Are there any DSD studies that have been previously published from the Kenyan setting for comparison purposes to the current study?

The study produced some interesting insights on recommendations for improvement of DSD. However, it would be useful if the authors could suggest some practical strategies to mitigate challenges raised such as privacy and preference for longer ART refills/less clinic visits.

One major limitation that the authors need to consider is the lack of a valid control group due to the before/after study design used. Therefore, there are some confounders that could not be adjusted for in the regression analysis such as ART policy changes (such as introduction of dolutegravir) and COVID-19 that would have had an impact on viral suppression and retention.

What are the strengths of this study?

Reviewer #2: Thank you for the opportunity to review the manuscript entitled “Effectiveness and experiences with differentiated service delivery of HIV care in Kisumu County, Kenya: A mixed methods study, 2014-2021”. Key strengths include the large sample size and duration of study observation. I also appreciate the mixed method approach and the results from the qualitative findings were insightful. My main concerns are methodologic; while DSD outcomes are notoriously difficult to estimate due to difficulty establishing a true counterfactual that can stand in as a comparison group for those enrolled in DSD models, the authors have not appropriately addressed this analytic challenge here. Much of the approach to definitions and data analysis is unclear from the manuscript text. In addition, the interpretation of the study findings which directly attribute the unusually high retention outcomes directly to the implementation of DSD needs addressing, especially given the noted potential for bias. I have several comments for the authors detailed below which should be address before publication could be considered:

Major comments:

1. Introduction – the expanse of existing DSD literature is not well covered. The statement “there is limited data on the effectiveness of DSD models as well as patient and provider perceptions of these approaches” seems unfounded – in fact, there are extensive publications around effectiveness and acceptability of DSD approaches in sub-saharan Africa (see authors Bolton-Moore, Bwire, Mukumbang, Mokhele, Huber, Pascoe, Long etc.) These should be explored and a more balanced perspective brought to the background.

2. The primary retention outcome seems problematic – currently it is defined as attendance at one visit in a 2-month window over the course of 12 or 24 months. This hardly seems like an accurate picture of retention in care. A client could simply have re-engaged in care and attended a visit in that period but missed several medication pickups during the course of the year. Given the data source available to the authors, alternate retention outcomes such as disengagement from care or treatment interruptions (missing at least one clinic visit by >28 days for example as per PEPFER definitions) should be considered.

3. The methods text describing person time in the analysis isn’t clear – what is meant by “people could enter the cohort at any time”? How were outcomes ascertained if a person hadn’t been on a DSD model for 12 months? Was transfer exclusively to another facility? How did decanting back to non-DSD model affect analysis? Were those included in outcome “retained”?

4. For the pre-baseline group, their person-time for the primary outcome needs to be better clarified. We assume they entered the cohort when they met eligibility criteria for DSD enrolment but when was their outcome defined? Was it 12 months after they met eligibility criteria?

5. The statistical methods are not wholly appropriate here. My main concern is with the use of logistic regression and odds ratios to present the key study findings – differences in risk between the three exposure groups. As the authors have the population denominator in these data, risk differences and relative risks with 95% CI can be calculated. The relative risk is what you are actually after in this scenario (probability of the event occurring in all exposed individuals versus the event occurring in all non-exposed individuals). Odds ratios tend to overestimate true relative risk with common outcomes; the overestimation becoming worse the more common the outcome (as in this case where the chosen outcome is >80% across all groups). This may be contributing to the very over-inflated OR estimates presented in the results and should be avoided in this case. I would suggest to rather estimate and present risk differences and relative risks (risk ratios) with 95% CI. You can then also use your confidence intervals to determine meaningful differences between groups rather than p values which should be reserved for hypothesis testing (also not what the authors are trying to achieve here).

6. Who is the group “imputed” to be on DSD? Is this the baseline group? If someone met the criteria for eligibility for DSD but wasn’t observed in the data to be in DSD care, that doesn’t mean they were. They could have simply refused to be in a DSD model and would be very different from those who opted to enrol?

7. The presentation of the outcomes seems incomplete. I have concerns with excluding those who elected not to be enrolled in DSD from retention calculations. The comparison group for the “baseline” include all who were eligible for DSD whether they would have enrolled or not. The same population should be included or at least presented for all groups – currently the outcomes for those who elected not to be in DSD are not shown.

8. Moreover, outcomes of those not retained should be clarified – numbers transferred, deceased, dropped out of care, or became unstable. I think it’s in Figure 2 but its so illegible, it could be helpful in the text.

9. The sensitivity analysis results need further explanation – in the methods they are indicated that these analyses would be to justify the inclusion of the ‘imputed” DSD group into the noted DSD group, yet no outcomes are presented for this group and the concept of a gap between clinical encounters of more than 6 months is described in the results (has never been reported or described prior to this point in the manuscript). This should be explained and justified in the methods.

10. The viral load suppression results presented were very limited. There is no text describing which VL tests were included and over which time periods in the methods nor any information on how missing VL test results were handled. Current interest in low-grade viremia offers an opportunity for some interesting analyses if the definition of suppressed VL (currently defined at <1000 copies/mL) were further stratified into 1) VL suppressed (<50 copies/mL), 2) low-grade viremia (50-1000 copies/mL) and 3) high grade viremia (>1000 copies/mL).

11. The text describing the multivariable analysis could use some revision for clarity; particularly with respect to which group and time period is being used as the reference group.

12. The focus group discussion results offer helpful insights and are a valuable contribution to the manuscript. They are however, somewhat one-sided in focusing on the benefits of the DSD models among those using them. The insights among those who were eligible but elected not to enroll in DSD models should also be presented; these insights are key to understanding challenges with uptake or implementation of DSD.

13. One of my key concerns is that the interpretation of the effect of DSD models on retention outcomes needs to be tempered. Retention rates are unusually high and direct comparisons with other published estimates from the region are missing from this manuscript. Currently, the text attributes the improvement in retention outcomes directly to DSD implementation. However, the authors note an important source of bias in this regard in the discussion – the program-wide intensified retention strategies carried out during this evaluation period certainly would have impacted on estimates of retention and create further problems with finding a true comparison group for the DSD enrolled group. This is not addressed in the limitations section at all. Sensitivity analyses could explore the impact this may be having but the sensitivity analyses presented are limited and confusing as to what they are trying to achieve.

14. The VL suppression outcomes have a similar problem in terms of attributing the improvements to DSD while other factors (DTG rollout, all the retention interventions introduced etc) would have made an important impact on numbers suppressed.

Minor comments:

1. Page 3 line 62: Sentence does not make sense – “Poor retention in HIV care and treatment programs has been identified as the most common reason for treatment failure among PLHIV on ART (14,15), with losses to follow up being a key driver of poor retention.” Loss to follow up is the same as poor retention? And not being on treatment could only be considered treatment failure surely?

2. “On the health care worker side, more time focused on managing complex cases and bolstering quality of care likely not only benefited clients enrolled in DSD models but also outcomes clinic wide” – could you expand on what the clinic-wide outcomes would be that would improve?

3. The behavioral theories noted in the discussion are great – these should be cited though and examples cited of this in action in other studies examining health outcomes to illustrate how this applies to DSD implementation.

4. The figures (especially figure 2) are almost completely illegible and make understanding results very difficult.

5. Suppl table multivariate outcomes need to report N achieving each outcome by each stratification level.

**Do you want your identity to be public for this peer review?** For information about this choice, including consent withdrawal, please see our Privacy Policy

Reviewer #1: No

Reviewer #2: No

---

## [Decision Letter · Decision Letter 1]

Effectiveness and experiences with differentiated service delivery of HIV care in Kisumu County, Kenya: A mixed methods study, 2014-2021

PGPH-D-24-02789R1

Dear Mr Mando,

We are pleased to inform you that your manuscript 'Effectiveness and experiences with differentiated service delivery of HIV care in Kisumu County, Kenya: A mixed methods study, 2014-2021' has been provisionally accepted for publication in PLOS Global Public Health.

Best regards,

Joel Msafiri Francis, MD, MS, PhD

Academic Editor

Reviewer Comments (if any, and for reference):

Reviewer's Responses to Questions

**Comments to the Author**

Reviewer #1: All comments have been addressed

publication criteria?

Reviewer #1: Yes

3. Has the statistical analysis been performed appropriately and rigorously?

Reviewer #1: Yes

4. Have the authors made all data underlying the findings in their manuscript fully available (please refer to the Data Availability Statement at the start of the manuscript PDF file)?

Reviewer #1: Yes

5. Is the manuscript presented in an intelligible fashion and written in standard English?

Reviewer #1: Yes

Reviewer #1: (No Response)

**Do you want your identity to be public for this peer review?** For information about this choice, including consent withdrawal, please see our Privacy Policy

Reviewer #1: No
